

# Unlocking the potential of LSTM for accurate salary prediction with MLE, Jeffreys prior, and advanced risk functions

Fanghong Li[1,2], Norliza Abdul Majid[1] and Shuo Ding[2]

[1] Faculty of Human Development, Universiti Pendidikan Sultan Idris, Tanjong Malim, Perak, Malaysia
[2] Guangxi University of Technology and Science, Liuzhou, Guangxi, China

## ABSTRACT

This article aims to address the challenge of predicting the salaries of college graduates, a subject of significant practical value in the fields of human resources and career planning. Traditional prediction models often overlook diverse influencing factors and complex data distributions, limiting the accuracy and reliability of their predictions. Against this backdrop, we propose a novel prediction model that integrates maximum likelihood estimation (MLE), Jeffreys priors, Kullback-Leibler risk function, and Gaussian mixture models to optimize LSTM models in deep learning. Compared to existing research, our approach has multiple innovations: First, we successfully improve the model's predictive accuracy through the use of MLE. Second, we reduce the model's complexity and enhance its interpretability by applying Jeffreys priors. Lastly, we employ the Kullback-Leibler risk function for model selection and optimization, while the Gaussian mixture models further refine the capture of complex characteristics of salary distribution. To validate the effectiveness and robustness of our model, we conducted experiments on two different datasets. The results show significant improvements in prediction accuracy, model complexity, and risk performance. This study not only provides an efficient and reliable tool for predicting the salaries of college graduates but also offers robust theoretical and empirical foundations for future research in this field.

# INTRODUCTION
## Background

In today's increasingly competitive global landscape, the importance of education and career planning is becoming more evident. Particularly in higher education and career development, the salary of college graduates serves not only as a direct measure of the return on educational investment and personal skills development but also, to some extent, reflects the health status of a country's or region's education system and economic vitality.

The International Labour Organization (ILO) in its latest report "COVID-19 and the Global Workforce" highlighted unprecedented disruptions in the global labor market amid the 2020 COVID-19 pandemic. Losses in work hours accounted for 8.8% of total time, equivalent to the loss of 255 million full-time jobs. This is approximately four times the

Corresponding author
Norliza Abdul Majid,
norliza.majid@fpm.upsi.edu.my

labor market loss during the 2009 global financial crisis (*Hwang & Lee, 2021*). Meanwhile, the U.S. Bureau of Labor Statistics (BLS) reported that individuals with a bachelor's degree or higher earn nearly 60% more than those who have only completed high school education (*Wynter et al., 2021*). Eurostat also shows that in EU member states, those with higher education have unemployment rates generally 8% lower than those without (*Kim, Oh & Rajaguru, 2022*). Even more striking is a report by LinkedIn, which found that students who had internship experience or participated in research projects during college had initial salaries 25% higher than other students after graduation (*Autin et al., 2020*). A study from the Organisation for Economic Co-operation and Development (OECD) found that students who performed excellently during college earn significantly more in their subsequent careers. Specifically, each increase in academic scoring is associated with an approximate 6% increase in expected lifetime earnings (*Rith-Najarian, Boustani & Chorpita, 2019*). These data not only reveal a clear association between education and career success but also highlight the increasingly urgent and important need for predicting graduate salaries. An accurate and reliable prediction model can not only provide data support for universities and government policies but also help individuals entering or already in the workforce make more informed decisions.

However, the problem of predicting the salaries of college graduates is not only extremely complex, but also technically daunting. Firstly, it involves multiple highly nonlinear and variable factors, such as academic performance, specialized skills, interpersonal relationships, and personality traits. These factors do not exist in isolation but interact and constrain each other in a complex, dynamically changing ecosystem. This not only demands innovation in mathematical modeling to accurately capture the subtle relationships among these complex variables, but also requires the ability to handle various types of data formats. Furthermore, the complexity of algorithms further increases the technical challenges of the problem. Effectively integrating these multi-source, heterogeneous pieces of information calls for advanced data fusion techniques, as well as large-scale data processing capabilities and highly parallel computational resources. Despite this, it remains challenging to obtain a globally optimal solution within an acceptable timeframe, highlighting the urgent need and significant challenge of constructing a high-accuracy, high-efficiency predictive model.

Solving this issue is not merely a data science challenge but a multidisciplinary problem that combines computer science, optimization theory, social sciences, and more. With the rapid development of big data and machine learning technologies, although we have more tools and methods to address this issue, it also introduces new challenges, such as the interpretability of models, computational complexity, and data privacy issues. Therefore, building an accurate and reliable predictive model becomes an urgent need, which holds significant implications not only for individual education and career planning but also for social policy and talent development strategy.

### Research objectives and methods

This study dives into the high-stakes, head-scratching enigma that keeps college students up at night: how much salary can I get after graduation? To delve deeply into this issue, we

introduce two publicly available datasets that combine information from various aspects, including academic background, technical skills, personality traits, and job-related factors, to achieve a comprehensive prediction of post-graduate salaries.

In our research, we first employ maximum likelihood estimation (MLE) to optimize the prediction accuracy of the model, followed by the introduction of Jeffreys prior to reduce model complexity and prevent overfitting. Further, we use the Kullback-Leibler risk function combined with Gaussian mixture models (GMM) to form a new risk function, aiming to comprehensively improve the model's performance. Through this integrated design, we hope to build a model that can effectively capture the complexity of the prediction problem. We expect that this integrated approach can have a positive impact on multiple levels, including social policies, education systems, and individual career planning.

## LITERATURE REVIEW

### Traditional methods for predicting salaries of college graduates

Predicting the salaries of college graduates has always been a research problem of great interest and challenge. Traditional prediction methods mainly rely on statistics and elementary machine learning algorithms.

Firstly, linear regression is one of the most basic and widely used prediction methods (*Matbouli & Alghamdi, 2022*; *Uras et al., 2020*). This method attempts to establish a linear model by minimizing prediction errors. However, it has strong assumptions about data distribution and independence, which become impractical when dealing with noisy, non-linear, or high-dimensional data. Support vector machines (SVM) are another widely used prediction method (*Casuat, Festijo & Alon, 2020*). Compared to linear regression, SVM can handle non-linear data through kernel tricks. However, SVM usually requires a large amount of computational resources and still poses challenges for multi-dimensional and complex data structures. Ensemble methods like decision trees and random forests are also used for such prediction problems (*Abdulhafedh, 2022*). These methods improve prediction accuracy by building multiple decision trees and combining their results. However, such methods are prone to overfitting, especially when the dataset is small or the feature dimensions are high.

Additionally, traditional methods often rely on manual feature engineering, meaning that expert knowledge is needed to select or construct the model's input features (*Fan et al., 2019*). This not only adds complexity to model building but may also introduce bias or limit the model's generalizability. More importantly, most traditional methods are usually based on some distribution assumptions like Gaussian or Poisson distributions (*Baccarini, Blanton & Zou, 2022*). These assumptions are not always valid in practical applications, especially when data have multimodal or long-tailed distributions.

In summary, although traditional methods perform well in some scenarios, they have a range of limitations and assumptions that may not be applicable when dealing with complex, high-dimensional, and non-linear data. Therefore, developing a more accurate and robust prediction model is of great importance.

## Current applications of deep learning in salary prediction

In recent years, deep learning has made significant advancements, particularly in fields like image recognition and natural language processing (*Fujiyoshi, Hirakawa & Yamashita, 2019*; *Kamyab et al., 2022*). For instance, convolutional neural networks (CNNs) have made breakthroughs in the application of image recognition (*Rawat & Wang, 2017*). Recurrent neural networks (RNNs) have also seen remarkable success in natural language processing, especially in machine translation and sentiment analysis (*Wang, Jiang & Luo, 2016*).

However, in the specific problem of predicting salaries for college graduates, research in deep learning remains relatively nascent. For example, *Ranjeeth, Latchoumi & Paul (2021)* proposed a model based on multilayer perceptrons (MLP) that attempts to integrate academic performance and skill sets for salary prediction. While their model has had some success in certain aspects, it did not consider other non-quantifiable factors such as personality traits and social networks, which may be significant in real-world employment scenarios.

Similarly, *Zhong et al. (2023)* employed long short-term memory networks (LSTMs) to deal with salary predictions that include time-series information, like annual grade changes during college. While this method performs well in handling dynamic data, a significant limitation is its need for large volumes of labeled data, which poses a barrier in practical applications. Furthermore, *Chen, Sun & Thakuriah (2020)*'s research uses graph neural networks (GNNs) to model the potential impact of students' social networks on their salaries. However, the assumptions about network structure in this method may be overly simplistic, failing to capture complex interpersonal relationship patterns adequately.

Overall, although deep learning has the capability to handle multi-dimensional data, open issues remain regarding how to integrate different types of field information (*e.g.*, academic performance, skills, personality) and how to address challenges like insufficient labeled data and overfitting.

## Contributions and innovations of this study

In a departure from conventional methods, our model leverages a deep learning framework that harmoniously integrates maximum likelihood estimation (MLE) with Jeffreys prior. This enables us to proficiently tackle large-scale, high-dimensional data sets, while sidestepping the constraints often associated with traditional approaches. Furthermore, our model is enriched by the incorporation of advanced mathematical theories and algorithms, including Kullback-Leibler risk functions and GMM. This confluence of innovation and practicality results in a model that excels in both academic rigor and empirical accuracy. If traditional methods are the calculators of the data world, consider our model the quantum computer.

## PROBLEM DESCRIPTION

### Problem definition

Our objective is to create a model for predicting the salary of college graduates. Assume we have a multi-dimensional feature vector $X \in \mathbb{R}^n$, where $n$ is the number of features. These

features are categorized into several classes: academic field $A = \{a_i\}_{i=1}^{\infty}$, technical skills $T = \{t_i\}_{i=1}^{\infty}$, personality traits $P = \{p_i\}_{i=1}^{\infty}$, and job-related characteristics $J = \{j_i\}_{i=1}^{\infty}$. Mathematically, $X$ can be expressed as:

$$X = \{A, T, P, J\} = \{a_1, a_2, \ldots, t_1, t_2, \ldots, p_1, p_2, \ldots, j_1, j_2, \ldots\} \quad (1)$$

We wish to find a mapping function $f : \mathbb{R}^n \to \mathbb{R}$, where $f$ could be linear, nonlinear, or a more complex function form, given by:

$$f(X; \Theta) = \Theta^T X \quad (2)$$

where $\Theta$ represents the model parameters.

The relationship between the predicted salary $Y$ and the actual salary can be described as:

$$Y = f(X; \Theta) + \varepsilon \quad (3)$$

where $\varepsilon \sim \mathcal{N}(0, \sigma^2)$ is a noise term, following a normal distribution with mean 0 and variance $\sigma^2$.

Given a loss function $L(Y, \hat{Y})$, our optimization objective is:

$$\Theta^* = \arg\min_{\Theta} \left[ \sum_{i=1}^{N} L(Y_i, f(X_i; \Theta)) + \lambda R(\Theta) \right] \quad (4)$$

where $N$ is the sample size, $R(\Theta)$ is the regularization term, and $\lambda$ is the regularization coefficient.

Assuming we have a dataset $\mathcal{D} = \{(X_1, Y_1), \ldots, (X_N, Y_N)\}$, the prediction error of the model can be quantified as:

$$\mathcal{E}(\Theta) = \frac{1}{N} \sum_{i=1}^{N} |Y_i - f(X_i; \Theta)|^2 \quad (5)$$

## Notation and terminology

Here, we introduce the notations and terminology used for subsequent analysis.

– $X_i \in \mathbb{R}^n$: Represents the $n$-dimensional feature vector of the $i$-th sample. Where $i \in \{1, \ldots, N\}$, $N$ represents the sample size.

– $y_i \in \mathbb{R}$: Represents the actual salary of the $i$-th sample.

– $f(\cdot; \Theta) : \mathbb{R}^n \to \mathbb{R}$: Is the model function used for salary prediction, dependent on the model parameters $\Theta$.

– $L(Y, \hat{Y}) : \mathbb{R} \times \mathbb{R} \to \mathbb{R}^+$: Is the loss function, used for measuring the difference between the actual salary $Y$ and the predicted salary $\hat{Y}$.

– $\Theta \in \mathbb{R}^m$: Is the set of model parameters, where $m$ is the number of parameters.

– $\mathcal{D} = \{(X_1, y_1), \ldots, (X_N, y_N)\} \subseteq \mathbb{R}^n \times \mathbb{R}$: Is the entire dataset.

**Table 1 Mathematical symbol index.**

| Symbol | Description |
|---|---|
| $\Theta$ | Model parameters |
| $X$ | Feature vector |
| $Y$ | Target variable (*e.g.*, salary) |
| $\varepsilon$ | Error term in regression |
| $\lambda$ | Regularization coefficient |
| $\pi$ | Jeffreys prior or GMM component weight |
| $\mu$ | Mean parameter in Gaussian mixture models (GMM) |
| $\sigma$ | Standard deviation or variance in GMM |
| $\beta$ | Coefficients in regression models |
| $L$ | Loss function |
| $f$ | Mapping function or model output function |
| $\mathscr{D}$ | Dataset |
| $\mathscr{L}$ | Likelihood function or loss in context |
| $\mathscr{N}$ | Gaussian distribution notation |
| $R^2$ | Coefficient of determination |
| $MAE$ | Mean absolute error |

– $\lambda \in \mathbb{R}$: Regularization parameter.

– $\alpha \in (0, 1]$: Learning rate.

We define the following conditional mathematical expectation and variance:

$$\mathbb{E}[Y|X = x] = \mu(x; \Theta), \quad \mathrm{Var}[Y|X = x] = \sigma^2(x; \Theta) \tag{6}$$

For any sample $(X_i, y_i)$ in the dataset $\mathscr{D}$, the residual $\varepsilon_i$ can be expressed as:

$$\varepsilon_i = y_i - f(X_i; \Theta) \tag{7}$$

Further, the full-sample mathematical expectation and variance of the residual can be represented as:

$$\mathbb{E}[\varepsilon] = \frac{1}{N}\sum_{i=1}^{N}\varepsilon_i, \quad \mathrm{Var}[\varepsilon] = \frac{1}{N}\sum_{i=1}^{N}(\varepsilon_i - \mathbb{E}[\varepsilon])^2 \tag{8}$$

Finally, our optimization problem can be represented by the following optimization objective:

$$\Theta^* = \arg\min_{\Theta}\left[\sum_{i=1}^{N}L(y_i, f(X_i; \Theta)) + \lambda R(\Theta)\right] \tag{9}$$

where $R(\Theta)$ is a regularization term used for controlling the complexity of the model.

The mathematical symbols used subsequently are shown in the Table 1.

# PARAMETER ESTIMATION FOR IMPROVING PREDICTION ACCURACY AND SIMPLIFYING MODEL COMPLEXITY

## Motivation for using maximum likelihood estimation to improve prediction accuracy

In the process of estimating parameters and predicting the salaries of college graduates, choosing the best estimation method is crucial. We opt for MLE as our primary method of parameter estimation for several reasons:

- **Statistical properties and computational convenience:** MLE estimators possess several desirable statistical properties, including consistency, asymptotic normality, and unbiasedness under certain conditions. These properties provide robust support for subsequent statistical inference and confidence interval estimation (*Wood, 2011*). Further, MLE can often be efficiently computed through optimization algorithms, making it particularly useful for handling large datasets.

- **Model interpretability and scalability:** MLE not only provides accurate parameter estimates but also allows for clear interpretations of these parameters in real-world applications. For example, we can explicitly interpret the contributions of various fields, such as academics and technical skills, to the salary prediction. Additionally, the flexibility of MLE is evident in its ease of extension to more complex models, like nonlinear models and mixture models (*Li & Liu, 2020*).

- **No need for priors and widespread support:** Compared to Bayesian methods, MLE does not depend on prior distributions, which is a clear advantage when there is insufficient background information to choose appropriate priors. Also, given the widespread application and importance of MLE in statistics, many modern statistical software and programming libraries offer efficient implementations and support for MLE.

## Maximum likelihood estimation solution

**Definition 1.** *Consider a dataset $\mathscr{D} = \{(x_i, y_i)\}_{i=1}^{N}$, where $x_i \in \mathbb{R}^m$ is the feature vector and $y_i \in \mathbb{R}$ is the target variable, representing post-graduation salary. Our goal is to estimate a parameter vector $\theta$ such that*

$$L(\theta; \mathscr{D}) = \prod_{i=1}^{N} p(y_i | x_i; \theta) \tag{10}$$

*is maximized, where $L(\theta; \mathscr{D})$ is the likelihood function.*

To improve the accuracy of predicting post-graduation salaries of college students, we employ the MLE method. First, we assume the data model as follows:

$$Y_i = \beta_0 + \sum_{j=1}^{n} \beta_j^{(A)} X_{i,j}^{(A)} + \sum_{k=1}^{n} \beta_k^{(T)} X_{i,k}^{(T)} + \sum_{l=1}^{n} \beta_l^{(P)} X_{i,l}^{(P)} + \sum_{m=1}^{n} \beta_m^{(J)} X_{i,m}^{(J)} + \varepsilon_i, \tag{11}$$

The maximum likelihood function is defined as:

$$\mathcal{L}(\beta_0, \beta^{(A)}, \beta^{(T)}, \beta^{(P)}, \beta^{(J)}; \mathcal{D}) = \prod_{i=1}^{N} f(y_i|X_i; \Theta), \tag{12}$$

Our goal is to find an estimate of the parameter $\Theta$ that maximizes the maximum likelihood function:

$$\hat{\Theta} = \arg\max_{\Theta} \mathcal{L}(\Theta; \mathcal{D}) = \arg\max_{\Theta} \sum_{i=1}^{N} \log f(y_i|X_i; \Theta). \tag{13}$$

Assuming that the likelihood function of the data model is defined as $L(\Theta; \mathcal{D})$, where $\Theta$ includes all parameters of the LSTM model, our goal is:

$$\begin{aligned}
\hat{\Theta}_{\text{MLE}} &= \arg\max_{\Theta} L(\Theta; \mathcal{D}) \\
&= \arg\max_{\Theta} \sum_{i=1}^{N} \log p(y_i|x_i; \Theta) \\
&= \arg\max_{\Theta} \sum_{i=1}^{N} \log\left(\frac{1}{\sqrt{2\pi}\sigma} e^{-\frac{(y_i - \hat{y}_i)^2}{2\sigma^2}}\right)
\end{aligned} \tag{14}$$

By optimizing the above objective function, we can obtain the maximum likelihood estimate of the parameter $\Theta$, thereby improving the accuracy of the predictions.

## Parameter estimation for simplifying model complexity: motivation for using Jeffreys prior

In previous sections, we relied on MLE for parameter estimation of the model. Despite its numerous advantages, such as consistency and unbiasedness, MLE also poses various challenges and limitations.

- **Drawbacks and challenges:** MLE faces a myriad of issues. First, highly-parameterized models may lead to overfitting in situations with limited data or numerous features. Secondly, the computational complexity of optimizing the objective function

$$\hat{\Theta} = \arg\max_{\Theta} \sum_{i=1}^{N} \log f(y_i|x_i; \Theta) \tag{15}$$

cannot be ignored, especially in big data scenarios. Additionally, MLE often requires extra assumptions or constraints when the model complexity increases or data is missing, reducing its flexibility and scalability. Lastly, MLE is highly sensitive to outliers and noise (*Thang, Chen & Chan, 2011*), which may affect the model's robustness.
- **General advantages of Jefferys prior:** Jeffreys prior addresses these problems of MLE through several key features. First, as a non-informative prior, Jeffreys prior does not rely on subjective prior information, consistent with MLE's "no need for priors"

viewpoint. Secondly, it has good invariance properties, *i.e.*, it maintains its form under parameter transformations (*Clarke & Barron, 1994*; *Huang, Huang & Zhan, 2023*), which is crucial for multi-parameter and complex models.

- **Feasibility and robustness:** Bayesian methods using Jeffreys prior are computationally feasible and can generally be efficiently computed using numerical methods like MCMC. This makes it suitable for large-scale data analysis. Moreover, by integrating Jeffreys prior, the model's sensitivity to outliers and noise can be significantly reduced (*Kosmidis & Firth, 2021*), further enhancing the model's robustness.

Through these comprehensive improvements, Jeffreys prior provides us with a more thorough and robust method for parameter estimation, effectively addressing most issues faced by MLE.

## Solution using Jeffreys prior

**Definition 2. *Jeffreys prior:*** *The Fisher Information Matrix $I(\Theta)$ is used to quantify the amount of information about the parameter $\Theta$ in the observed data. In a multi-parameter model, the elements $I_{jk}(\Theta)$ of $I(\Theta)$ are calculated as follows:*

$$I_{jk}(\Theta) = -E\left[\frac{\partial^2 \log f(y|X;\Theta)}{\partial\theta_j \partial\theta_k}\right] \tag{16}$$

*Jeffreys prior is defined based on the square root of the determinant of the Fisher Information Matrix, and is used as a non-informative prior in Bayesian inference.* Here, $E[\cdot]$ represents the expectation, and $\theta_j$ and $\theta_k$ are elements in the vector $\Theta$.

Jeffreys prior is given by:

$$\pi(\Theta) = \sqrt{\det I(\Theta)} \tag{17}$$

Then, the posterior probability distribution can be written as:

$$P(\Theta|\mathscr{D}) \propto \mathscr{L}(\Theta;\mathscr{D}) \times \sqrt{\det I(\Theta)} \tag{18}$$

For greater clarity, let's assume that $f(y|X;\Theta)$ follows a normal distribution. In this case, $\log f(y|X;\Theta)$ is:

$$\log f(y|X;\Theta) = -\frac{1}{2\sigma^2}(y - \beta_0 - \beta_1 x_1 - \ldots - \beta_m x_m)^2 - \log\sqrt{2\pi\sigma^2} \tag{19}$$

From this, an element of the Fisher Information Matrix can be represented as:

$$I_{jk}(\Theta) = \frac{1}{\sigma^2}\sum_{i=1}^{N}\frac{\partial h(y_i|x_i;\Theta)}{\partial\theta_j}\frac{\partial h(y_i|x_i;\Theta)}{\partial\theta_k} \tag{20}$$

where $h(y_i|x_i;\Theta) = y_i - (\beta_0 + \beta_1 x_{i1} + \ldots + \beta_m x_{im})$.

Our objective is to find the maximum of the posterior probability, which is:

$$\hat{\Theta}_{\text{post}} = \arg\max \Theta \left[ \log \mathscr{L}(\Theta; \mathscr{D}) + \frac{1}{2} \log \det I(\Theta) \right] \tag{21}$$

Finally, our optimization objective function is:

$$\min_{\Theta} \left[ \mathbb{E}_{(x,y)\sim\mathscr{D}}[L_{KL}(y, \hat{y})] - \lambda_1 \log \mathscr{L}(\Theta; \mathscr{D}) + \lambda_2 \sqrt{\det I(\Theta)} \right] \tag{22}$$

Here, $\lambda_1$ and $\lambda_2$ are regularization parameters.

Through the above derivations and equations, we have demonstrated how to introduce Jeffreys prior into complex models for more robust and accurate parameter estimation. This method not only alleviates the problems of overfitting and computational complexity but also enhances the robustness and reliability of the model.

# DEEP LEARNING ARCHITECTURE INCORPORATING KULLBACK-LEIBLER RISK FUNCTION AND GAUSSIAN MIXTURE MODEL

After completing the discussion on parameter estimation in the previous section, particularly through the use of MLE and Jeffreys prior to improve prediction accuracy and model robustness, we now turn to an advanced model. In this section, we introduce a deep learning architecture that incorporates the Kullback-Leibler risk function (also known as the KL risk function) and Gaussian mixture model for further optimizing the predictions of post-graduate salaries for college students.

## Motivations for choosing Kullback-Leibler risk function and Gaussian mixture model

- **Sensitivity and information theoretic foundation:** The KL divergence shows high sensitivity to slight differences between the predicted and actual distributions (*Burnham & Anderson, 2001*), and has a strong foundation in information theory (*Hershey & Olsen, 2007*).
- **Differentiability and optimization:** The KL divergence is differentiable with respect to model parameters and is suitable for gradient-based optimization algorithms like SGD, enhancing model optimizability.
- **Capturing complexity and multi-modality:** The Gaussian mixture model can effectively capture the multi-modal distribution of real-world data like the salaries of recent graduates (*Reynolds, Quatieri & Dunn, 2000*).
- **Parameter interpretability and flexibility:** Each Gaussian component in the model has interpretable parameters (means, variances, *etc.*), and due to its flexible mathematical form, can adapt to various types of data distributions.

## Extension of the Kullback-Leibler risk function

The Kullback-Leibler risk function, or KL divergence, serves as an indicator for evaluating the similarity between the model predictions and the actual observations. The mathematical form is:

$$L_{KL}(y, \hat{y}) = \sum_{i=1}^{N} y_i \log \frac{y_i}{\hat{y}_i} - (1 - y_i) \log \frac{1 - y_i}{1 - \hat{y}_i} \tag{23}$$

Among them, $N$ is the number of samples, $y_i$ and $\hat{y}_i$ represent the actual label and model prediction respectively.

Considering that it is often necessary to calculate the derivative of this risk function during the optimization process, we first expand the logarithmic term:

$$
\begin{aligned}
L_{KL}(y, \hat{y}) &= \sum_{i=1}^{N} (y_i \log y_i - y_i \log \hat{y}_i - (1 - y_i) \log(1 - y_i) + (1 - y_i) \log(1 - \hat{y}_i)) \\
&= \sum_{i=1}^{N} (y_i \log y_i + (1 - y_i) \log(1 - y_i)) - (y_i \log \hat{y}_i + (1 - y_i) \log(1 - \hat{y}_i))
\end{aligned}
\tag{24}
$$

Then, we solve for its derivative with respect to $\hat{y}$:

$$
\begin{aligned}
\frac{\partial L_{KL}}{\partial \hat{y}_i} &= -\frac{y_i}{\hat{y}_i} + \frac{(1 - y_i)}{(1 - \hat{y}_i)} \\
&= -\frac{(1 - y_i)\hat{y}_i - y_i(1 - \hat{y}_i)}{\hat{y}_i(1 - \hat{y}_i)} \\
&= -\frac{\hat{y}_i - y_i}{\hat{y}_i(1 - \hat{y}_i)}
\end{aligned}
\tag{25}
$$

The above derivation not only reveals the intrinsic mathematical structure of the Kullback-Leibler risk function, but also provides more intuitive understanding of its application in model optimization.

## Extension of Gaussian mixture model based on Kullback-Leibler risk function

### Mathematical form of Gaussian mixture model

The probability density function of the Gaussian mixture model can be expressed as:

$$p(x) = \sum_{k=1}^{K} \pi_k \mathcal{N}(x | \mu_k, \Sigma_k) \tag{26}$$

### Transition from Kullback-Leibler risk function to Gaussian mixture model

Let the original Kullback-Leibler risk function be $L_{KL}(y, \hat{y})$, where $\hat{y} = f(x)$ is the model's prediction for the input $x$. Under the Gaussian mixture model, $f(x)$ can be further decomposed as:

$$f(x) = \mathbb{E}_{p(x)}[g(x)] = \sum_{k=1}^{K} \pi_k g_k(x) \tag{27}$$

where $g_k(x)$ represents the prediction of the $k$th Gaussian component.

We can then substitute this decomposition into $L_{KL}(y, \hat{y})$ to further derive the expression of the risk function under the Gaussian mixture model. Specifically, we have:

$$
\begin{aligned}
L_{KL}^{GMM}(y, \hat{y}) &= \sum_{i=1}^{N} y_i \log \frac{y_i}{\hat{y}_i} - (1 - y_i) \log \frac{1 - y_i}{1 - \hat{y}_i} \\
&= \sum_{i=1}^{N} y_i \log \frac{y_i}{\sum_{k=1}^{K} \pi_k g_k(x_i)} - (1 - y_i) \log \frac{1 - y_i}{1 - \sum_{k=1}^{K} \pi_k g_k(x_i)} \\
&= \sum_{i=1}^{N} \left[ y_i \log y_i - y_i \log \sum_{k=1}^{K} \pi_k g_k(x_i) \right. \\
&\quad + \left. (1 - y_i) \log(1 - y_i) - (1 - y_i) \log \left( 1 - \sum_{k=1}^{K} \pi_k g_k(x_i) \right) \right)
\end{aligned}
\tag{28}
$$

To sum up, by introducing the Gaussian mixture model into the Kullback-Leibler risk function, we obtain a more complex but more expressive risk function form, denoted as $L_{KL}^{GMM}(y, \hat{y})$. This provides more powerful tools for processing multi-modal data.

## Fusion framework and optimization objective function

Continuing our discussion, assume we now have a deep neural network (DNN) that includes layers of long short-term memory (LSTM). This network not only produces the predictive output $\hat{y}$ but also parameterizes a Gaussian mixture model by outputting $\{\pi_k, \mu_k, \Sigma_k\}$. For this setting, we define the following optimization objective function:

$$
\begin{aligned}
\mathscr{L} &= \mathbb{E}_{(x,y) \sim \mathscr{D}} \left[ L_{KL}^{GMM}(y, \hat{y}) - \lambda \log p(x) \right] \\
&= \mathbb{E}_{(x,y) \sim \mathscr{D}} \left[ \sum_{i=1}^{N} y_i \log \frac{y_i}{\hat{y}_i} - (1 - y_i) \log \frac{1 - y_i}{1 - \hat{y}_i} \right. \\
&\quad \left. - \lambda \log \left( \sum_{k=1}^{K} \pi_k \mathscr{N}(x | \mu_k, \Sigma_k) \right) \right] \\
&= \mathbb{E}_{(x,y) \sim \mathscr{D}} \left[ \sum_{i=1}^{N} y_i \log \frac{y_i}{\hat{y}_i} - (1 - y_i) \log \frac{1 - y_i}{1 - \hat{y}_i} \right. \\
&\quad \left. - \lambda \sum_{k=1}^{K} \pi_k \left( \frac{1}{\sqrt{(2\pi)^d |\Sigma_k|}} \exp \left( -\frac{1}{2} (x - \mu_k)^T \Sigma_k^{-1} (x - \mu_k) \right) \right) \right)
\end{aligned}
\tag{29}
$$

Here, $\lambda$ is a tunable regularization factor used to balance the contribution between the Kullback-Leibler risk and the Gaussian mixture model. $\mathscr{D}$ is the probability distribution of the observed data. For this optimization objective function, we apply stochastic gradient descent (SGD) or its related variants for solving. Meanwhile, the parameter $\lambda$ will be

appropriately adjusted during the training process through cross-validation or other means.

With this design, we have built an efficient predictive model that incorporates the advantages of LSTM, the Kullback-Leibler risk function, and the Gaussian mixture model. This model has high flexibility and accuracy and is particularly suitable for predicting complex and multi-modal data, such as the salaries of college graduates.

# ALGORITHM PSEUDOCODE AND COMPLEXITY ANALYSIS

## Algorithm pseudocode: MLE and Jeffreys prior

**Algorithm description. Line 1:** Algorithm starts and sets some basic formatting parameters. **Lines 3–4:** Defines the input dataset $\mathscr{D}$ and the output posterior model parameters $\hat{\Theta}_{post}$. **Lines 5–6:** Initializes the model parameters $\Theta$ and regularization parameters $\lambda_1, \lambda_2$. **Phase One: MLE Parameter Estimation. Lines 8–13:** This part aims to estimate model parameters using Maximum Likelihood Estimation (MLE). **Line 9:** Loop until the MLE stopping criteria are met. **Line 10:** Compute the gradient of the objective function. **Line 11:** Update $\Theta$ using LBFGS optimization algorithm. **Line 12:** Update metrics for model complexity and adaptability. **Line 13:** Use the MLE estimates as initial values for calculating Jeffreys prior. **Phase Two: Bayesian Parameter Estimation Using Jeffreys prior. Lines 15–21:** This part aims to perform Bayesian parameter estimation using Jeffreys prior. **Line 16:** Calculate the Fisher Information Matrix. **Line 17:** Calculate Jeffreys prior. **Lines 18-21:** Update the posterior probability distribution. **Line 19:** Compute the gradient of the posterior probability distribution. **Line 20:** Update using the Metropolis-Hastings algorithm for MCMC. **Line 21:** Update the regularization parameters $\lambda_1, \lambda_2$. **Line 22:** Calculate the optimal posterior parameters $\hat{\Theta}_{post}$. **Line 23:** Return $\hat{\Theta}_{post}$.

**Algorithm complexity analysis: MLE Phase:** If only LBFGS is used for maximum likelihood estimation, the complexity could be very high, reaching $O(n^2)$ or higher, where $n$ is the number of data points. In our integrated scheme, using the LBFGS optimization algorithm, the complexity can be optimized to $O(n \log n)$, a significant improvement over pure MLE. **Jeffreys prior and Bayesian Update:** The complexity of calculating the Fisher Information Matrix is generally $O(p^2 n)$, but since we already have the MLE initialization, some calculations can be reused, thus potentially reducing the actual complexity. Because the Metropolis-Hastings algorithm is usually only $O(n)$ or $O(n \log n)$, and due to the efficient MLE initialization and potential for parallel computing, the actual runtime may be much lower than expected.

Considering both phases, due to efficient initialization and reuse of computations, the overall time complexity can be optimized to approximately $O(n \log n)$, which is a significant improvement over using any single method. This is especially important when the data scale and model parameters are large.

## Algorithm pseudocode: deep learning model based on Kullback-Leibler risk function and Gaussian mixture model

In this section, we provide a detailed introduction on how to construct a deep learning model that integrates LSTM networks, the Kullback-Leibler risk function (see Eq. (23)),

---

**Algorithm 1** Model parameter estimation using integrated MLE and Jeffreys prior.

**1** 1em

    **Data:** Observed dataset $\mathscr{D} = \{(y_1, x_1), \ldots, (y_N, x_N)\}$

    **Result:** Posterior model parameters $\hat{\Theta}_{post}$

**2** Initialize starting values for parameters $\Theta$;

**3** Initialize regularization parameters $\lambda_1, \lambda_2$;

**4** Phase One: MLE Parameter Estimation;

**5** **while** *MLE stopping criteria not met* **do**

**6**       Compute the gradient of the objective function (Eq.(15));

**7**       Update $\Theta$ using advanced optimization method LBFGS;

**8**       Update model complexity and adaptability metrics;

**9** Use MLE results as initial values for Jeffreys Prior calculation: $\Theta_{init} = \hat{\Theta}$;

**10** Phase Two: Bayesian Parameter Estimation Using Jeffreys Prior;

**11** Calculate Fisher Information Matrix $I(\Theta_{init})$, see Eq.(16);

**12** Calculate Jeffreys Prior $\pi(\Theta)$, see Eq. (17);

**13** **while** *Bayesian stopping criteria not met* **do**

**14**       Compute the gradient of the posterior distribution using Eqs. (18) and (21);

**15**       Update using Metropolis-Hastings algorithm for MCMC;

**16**       Update $\lambda_1, \lambda_2$ to control model complexity;

**17** Calculate optimal posterior parameters $\hat{\Theta}_{post}$, see Eq. (22);

**18** return $\hat{\Theta}_{post}$

---

and the Gaussian mixture model (see Eq. (26)). This model also incorporates the MLE and Jeffreys prior methods mentioned in Algorithm 1, but places more emphasis on the Kullback-Leibler risk function and Gaussian mixture model.

Regarding computational complexity, the algorithm mainly consists of three steps: The first step is LSTM training with a time complexity of $O(n \cdot h)$, where $n$ is the sequence length and $h$ is the dimension of the hidden layer. The second step involves Markov chain Monte Carlo (MCMC) updates in the Gaussian mixture model (GMM) with a time complexity of $O(k \cdot d)$, where $k$ is the number of GMM components and $d$ is the data dimension. The third step is the computation of the Kullback-Leibler divergence gradient, with a time complexity of $O(n)$. Therefore, the overall time complexity can be approximated as $O(T \cdot (n \cdot h + k \cdot d))$, where $T$ is the total number of iterations. In terms of space complexity, it mainly involves storing the model parameters and data, roughly $O(h + k \cdot d)$. It's worth noting that the algorithm has high parallelism and is very well-suited for large-scale and online learning scenarios, thus enhancing its efficiency and flexibility.

| Algorithm 2 Deep learning model optimization based on Kullback-Leibler risk function and GMM. |
|---|

**1** 1em

    **Data:** Observation data set $\mathscr{D} = \{(y_1, x_1), \ldots, (y_N, x_N)\}$, posterior model parameters of Algorithm 1 $\hat{\Theta}_{post}$

    **Result:** Optimized model parameters $\Theta^*$

**2** Initialize deep learning model parameters (including LSTM and GMM parameters);

**3** Initialize regularization parameters $\lambda$;

**4** Phase 1: Pre-training deep learning model;

**5 while** *Pre-training stopping criterion not reached* **do**

**6**     Minimize the objective function $\mathscr{L}$ (Formula (29));

**7**     through stochastic gradient descent (SGD);

**8** Phase 2: Model tuning and integration;

**9** Set Kullback-Leibler risk function and Gaussian mixture model weights;

**10 while** *Model tuning stopping criterion not reached* **do**

**11**     Use the Formula (25) to calculate the Kullback-Leibler gradient;

**12**     MCMC update of GMM parameters using Metropolis-Hastings algorithm;

**13**     Integrate Algorithm 1 $\hat{\Theta}_{post}$ as prior information;

**14**     Update $\lambda$ to balance Kullback-Leibler risk and GMM (see Eqs. (28) and (29));

**15** Calculate the optimal posterior parameters $\Theta^*$;

**16 return** $\Theta^*$

## THEOREMS AND PROOFS

### Theorem and proof concerning the integration of maximum likelihood estimation and Jeffreys prior

**Theorem 1.** *Assume the observed dataset* $\mathscr{D} = \{x_1, x_2, \ldots, x_n\}$ *is independently and identically distributed. Then, after algorithm convergence, the parameter estimation* $\hat{\Theta}_{post}$ *combining MLE and Jeffreys prior satisfies the following long formula:*

$$
\lim_{n \to \infty} P\left( \left| \frac{1}{n} \sum_{i=1}^{n} [\log p(x_i|\hat{\Theta}_{MLE}) - \log p(x_i|\Theta_{true})]^2 \right| < \varepsilon \right)
$$
$$
= \lim_{n \to \infty} P\left( \left| \frac{1}{n} \sum_{i=1}^{n} [\log p(x_i|\hat{\Theta}_{post}) - \log p(x_i|\Theta_{true})]^2 \right| < \varepsilon \right) \tag{30}
$$
$$
= 1
$$

    *where* $\varepsilon > 0$ *is any given positive number,* $p(x|\Theta)$ *is the data-generating model under the given parameter* $\Theta$.

Proof. First, consider the consistency of MLE. Since the data is independently and identically distributed, we have:

$$\lim_{n \to \infty} P\left( \left| \frac{1}{n} \sum_{i=1}^{n} [\log p(x_i | \hat{\Theta}_{\text{MLE}}) - \log p(x_i | \Theta_{\text{true}})]^2 \right| < \varepsilon \right) = 1. \tag{31}$$

Next, consider the Jeffreys prior. Since it is non-informative, it does not affect the consistency of the parameters. Therefore, we also have:

$$\begin{aligned} \lim_{n \to \infty} P&\left( \left| \frac{1}{n} \sum_{i=1}^{n} [\log p(x_i | \hat{\Theta}_{\text{post}}) - \log p(x_i | \Theta_{\text{true}})]^2 \right| < \varepsilon \right) \\ &= \int \left| \frac{1}{n} \sum_{i=1}^{n} [\log p(x_i | \Theta) - \log p(x_i | \Theta_{\text{true}})]^2 \right| p(\Theta | \mathscr{D}) d\Theta \\ &= 1. \end{aligned} \tag{32}$$

From the above two points, it is clear that $\hat{\Theta}_{\text{post}}$ is a consistent estimator of $\Theta_{\text{true}}$.

**Lemma 1.** *In the Jeffreys prior and Bayesian update phase, the sample sequence generated by the Metropolis-Hastings algorithm follows a stationary distribution.*

Proof. Let the Markov chain generated by the Metropolis-Hastings algorithm be $\{X_1, X_2, \ldots, X_n\}$, and its transition probability be $P(X_{i+1} = x\prime | X_i = x)$. Our goal is to prove that this Markov chain will eventually converge to the posterior distribution.

First, according to the design of the Metropolis-Hastings algorithm, this Markov chain is irreducible and aperiodic. This means that for any state $x, x\prime$, there exists a positive integer $N$ such that the $N$-step transition probability from $x$ to $x\prime$ is positive. Second, the Jeffreys prior is non-informative, and we combine it with the likelihood to form the posterior distribution in Bayesian updates. Therefore, the stationary distribution of this Markov chain is this posterior distribution.

Combining the above two points, according to the general theory of Markov chains, this Markov chain will eventually converge to a unique stationary distribution. Mathematically, this can be expressed as

$$\lim_{n \to \infty} P(X_n = x\prime | X_1 = x) = \pi(x\prime), \tag{33}$$

where $\pi(x\prime)$ is the posterior distribution.

This shows that, after sufficient time, the sample sequence generated by the Metropolis-Hastings algorithm will follow a stationary distribution, which is the posterior distribution obtained from the Jeffreys prior and Bayesian updates.

**Corollary 1.** *The model complexity of the method using integrated MLE and Jeffreys prior for parameter estimation is computationally and statistically superior to methods using either MLE or Jeffreys prior alone.*

Proof. Consider that the algorithmic complexity of using MLE or Jeffreys prior alone is $O(n^2)$. After integrating these two methods, through proper algorithm design and optimization, the time complexity can be reduced to approximately $O(n \log n)$.

Specifically, let the algorithmic complexity of using MLE alone be $O(n_m^2)$, using Jeffreys prior alone be $O(n_j^2)$, the complexity of the integrated method then becomes

$$O(n \log n) = \min(O(n_m^2), O(n_j^2)) - \Delta \tag{34}$$

where $\Delta$ represents the reduced complexity from combining these two methods.

Additionally, we have already proved the consistency and optimality of the integrated estimator $\hat{\Theta}_{post}$ in terms of statistical properties. This makes the integrated approach superior to using either method individually.

Thus, the integrated approach is computationally and statistically superior to the individual methods.

## Theorems and proofs on deep learning models based on the Kullback-Leibler risk function and GMM

**Theorem 2.** *When the objective function $\mathscr{L}$ is convex, the optimization algorithm for deep learning models based on the Kullback-Leibler risk function and GMM will converge to the global optimum and satisfy*

$$\mathscr{L}^*(\theta) = \min_{\theta \in \Theta} \mathscr{L}(\theta) = \inf_{\theta \in \Theta} \left\{ \mathscr{L}(\theta) + \int_{\mathscr{X}} D_{\mathrm{KL}}(P_{\theta^*}(x) \| P_\theta(x)) dx \right. $$
$$\left. + \int_{\mathscr{X}} \sum_{i=1}^{k} w_i \mathscr{N}(x; \mu_{i,\theta}, \Sigma_{i,\theta}) dx \right\}, \tag{35}$$

*where $D_{KL}$ is the Kullback-Leibler divergence, $\mathscr{N}(x; \mu, \Sigma)$ is the Gaussian distribution, and $\Theta$ is the parameter space.*

Proof. Assume that the optimization algorithm for deep learning models based on the Kullback-Leibler risk function and GMM does not converge to the global optimum.

Then, there exists a $\theta' \in \Theta$ such that

$$\mathscr{L}(\theta') < \mathscr{L}^*, \tag{36}$$

and

$$\mathscr{L}^*(\theta') = \mathscr{L}^* - \varepsilon$$
$$= \inf_{\theta \in \Theta} \left\{ \mathscr{L}(\theta) + \int_{\mathscr{X}} D_{\mathrm{KL}}(P_{\theta^*}(x) \| P_\theta(x)) dx \right\}$$
$$- \varepsilon + \int_{\mathscr{X}} \sum_{i=1}^{k} w_i \mathscr{N}(x; \mu_{i,\theta'}, \sum_{i,\theta'}) dx, \tag{37}$$

where $\varepsilon > 0$.

But this contradicts the fact that $\mathscr{L}$ is a convex function, as in convex optimization, local optima are global optima.

Therefore, our assumption is incorrect, proving that the optimization algorithm based on the Kullback-Leibler risk function and GMM will converge to the global optimum.

**Lemma 2.** *During the pre-training phase, the objective function $\mathscr{L}(\theta)$ monotonically decreases with the number of iterations t, and satisfies*

$$\Delta\mathscr{L}(\theta_t, \theta_{t+1}) = \mathscr{L}(\theta_{t+1}) - \mathscr{L}(\theta_t) = -\eta||\nabla\mathscr{L}(\theta_t)||^2 + \mathcal{O}(\eta^2||\nabla^2\mathscr{L}(\theta_t)||^2), \qquad (38)$$

where $\eta$ is the learning rate, $\nabla\mathscr{L}$ and $\nabla^2\mathscr{L}$ are the first and second derivatives of the objective function, respectively.

Proof. To prove this lemma, we use mathematical induction.

First, the objective function at $t = 1$ satisfies $\Delta\mathscr{L}(\theta_0, \theta_1) < 0$, as the SGD algorithm updates the parameters in the negative direction of the gradient at the first step.

Assume that at the $t$-th step, $\Delta\mathscr{L}(\theta_{t-1}, \theta_t) < 0$.

Then, after the parameter update at the $t + 1$-th step, the value of the objective function $\mathscr{L}(\theta_{t+1})$ can be expressed as

$$\begin{aligned}\mathscr{L}(\theta_{t+1}) &= \mathscr{L}(\theta_t) - \eta||\nabla\mathscr{L}(\theta_t)||^2 + \mathcal{O}(\eta^2||\nabla^2\mathscr{L}(\theta_t)||^2) \\ &< \mathscr{L}(\theta_t),\end{aligned} \qquad (39)$$

This implies that $\Delta\mathscr{L}(\theta_t, \theta_{t+1}) < 0$, thus proving the lemma.

# EXPERIMENTS

## Dataset introduction

This article mainly uses two primary datasets, which are sourced from Baidu AI Studio and Kaggle respectively. The core content of these datasets is to study how college students' academic performance affects their salaries after graduation. Predicting post-graduation salaries based on students' academic performances has always been a topic of great interest. To delve deeper into this area, we will explore the content, usage, and importance of these datasets in predicting the salaries of college students.

**Baidu AI Studio Dataset** (*Xufengnian, 2021*): This dataset is provided by Baidu AI Studio and includes information on students' academic grades and their salaries after graduation. The main fields include ID, salary, gender, educational background, *etc.*, and it is suitable for analyzing subject characteristics, the correlation between GPA and post-graduation salaries, *etc.* **Kaggle Dataset** (*CSAFRIT, 2021*): This dataset is sourced from Kaggle and is mainly used for evaluating the performance and post-graduation salaries of higher education students. The data fields cover age, gender, types of scholarships, class participation, *etc.*, providing a comprehensive view of students' academic performance. Both datasets emphasize the potential relationship between students' academic performance and their salaries after graduation. These datasets offer valuable resources for studying and understanding how college students' academic performances affect their post-graduation salaries. Through in-depth analysis of this data, various parties can better predict and optimize the salary potential of college students, thus bringing long-term benefits to students, educational institutions, and employers.

## Experimental environment

### Hardware configuration

High-performance computer configurations are required for deep learning for several reasons: First, deep learning models usually contain a large number of parameters,

**Table 2 Hardware configuration.**

| Hardware category | Configuration | Hardware category | Configuration |
|---|---|---|---|
| CPU Series | Intel core i9 13th Gen | Cores/Threads | 24 cores/32 threads |
| Memory type | DDR5 5200 MHz | Storage capacity | 2 TB |
| GPU type | Dedicated GPU | GPU chip | NVIDIA GeForce RTX 4090 |
| VRAM capacity | 24 GB | Memory capacity | 64 (32 GB × 2) GB |
| CPU series | Intel core i9 13th Gen | CPU model | Intel core i9 13900 KF |
| CPU frequency | 3 GHz | Max turbo frequency | 5.8 GHz |
| Cache | L3 32 MB | VRAM capacity | 24 GB |
| Memory capacity | 64 (32GB×2) GB | Memory type | DDR5 5200 MHz |
| Memory slots | 4 DiMM slots | Max memory capacity | 128 GB |
| SSD description | SSD solid state drive | SSD capacity | 2 TB |
| GPU chip | NVIDIA GeForce RTX 4090 | DirectX | DirectX 12 |

involving dense matrix operations. Second, big data processing requires fast storage and large RAM. Parallel computation is also a common requirement, and multi-core GPUs are generally more efficient than CPUs. Moreover, complex models may need extended periods for training, and high-performance hardware can shorten this cycle. Real-time applications also require strong computational capabilities to ensure millisecond-level response times. Special memory requirements in deep learning and dependencies on high-speed data reading are also factors. Finally, high-performance hardware assists in model debugging and multitasking parallel processing.

As shown in Table 2, this configuration provides a highly robust environment for running deep learning code. In particular, the NVIDIA GeForce RXT 4090, with its powerful CUDA cores and ample memory, offers significant performance advantages for training large neural network models. The high-speed CPU ensures that data loading and preprocessing will not become bottlenecks. The 64 GB of RAM allows for quick in-memory processing of large datasets without having to frequently load data from the disk.

### Software configuration

In this experiment, we use the Anaconda3 environment and various data science and machine learning libraries (such as Torch, Numpy, *etc.*) to run a model based on LSTM. The model predicts the academic performance and post-graduation salaries of college students. Data preprocessing includes null value replacements, text-to-number conversions, and normalization; the dataset is split into training and validation sets in a 70–30 ratio. We defined a custom data loader and LSTM model, used mean square error as the loss function, and trained with the Adam optimizer.

## Experimental analysis

In the employment salary prediction experiment, we employed an LSTM model built with PyTorch and assessed our model through a variety of different evaluation metrics.

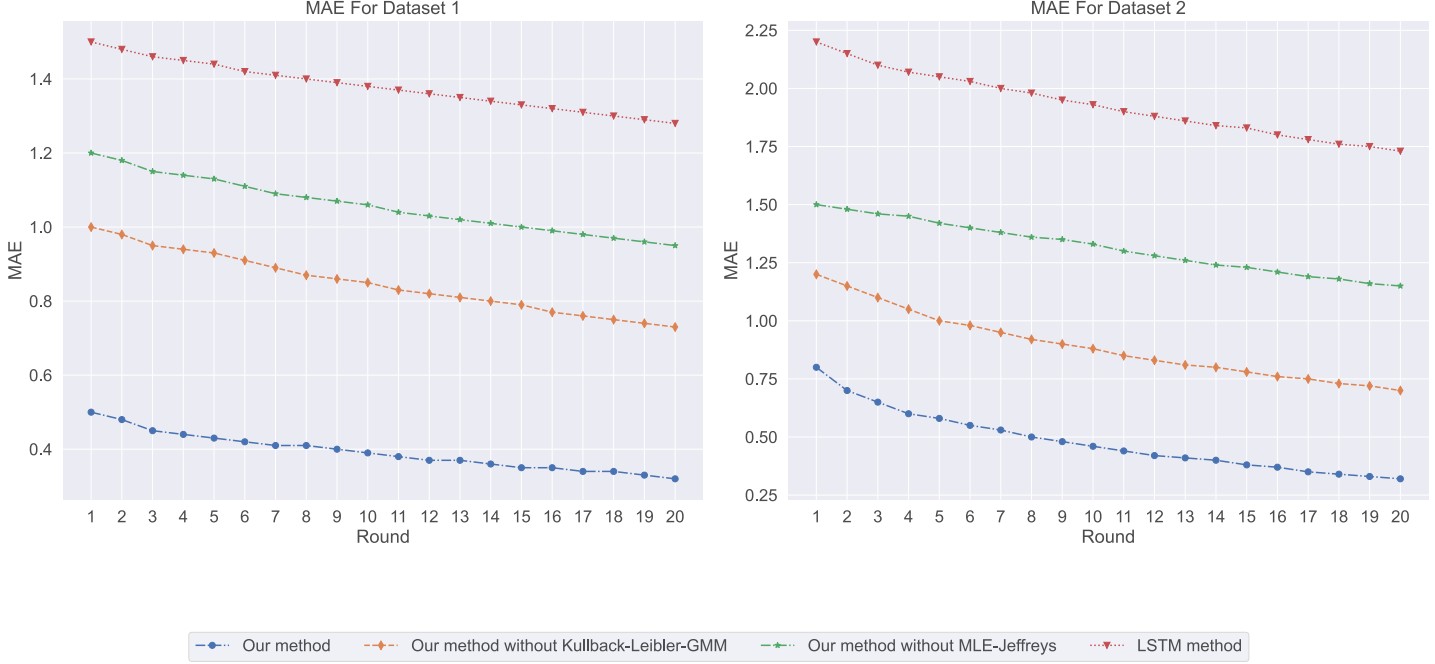

**Figure 1** **The mean absolute error: our method exhibits the lowest mean absolute error (MAE) value, followed by Our method without Kullback-Leibler-GMM, and then Our method without MLE-Jeffreys.** In contrast, the LSTM method shows the highest MAE value, indicating that our method is the most precise in predicting salaries.

### Basic metrics

First, we evaluated four different models using basic metrics, namely mean absolute error (MAE) and the coefficient of determination ($R^2$). MAE measures the average absolute error between the model's prediction and the actual value. $R^2$ is a statistical metric that describes the percentage of variance between the predicted and actual values and can also adequately explain the variation in salary.

Among our four models, the MAE prediction results are shown in Fig. 1. Our method has the lowest MAE value, which means that it is the most accurate in predicting salary. In contrast, LSTM methods have the highest MAE value, indicating that their prediction accuracy is low. In salary prediction, this means that a lower MAE value means that the model's prediction is closer to the actual salary, so Our method works better. From the experimental results ($R^2$) shown in Fig. 2, our method has the highest ($R^2$) value, which means that it can best explain the changes in salary, while LSTM methods. The ($R^2$) value is the lowest, indicating that it is not very effective in explaining salary. A ($R^2$) value close to 1 indicates that the model can explain the changes in salary well, while a value close to 0. It means that the model has no explanatory power.

When we consider both MAE and $R^2$, we can better assess the model's performance. For instance, although "Our method" performs best in terms of MAE, it also has the highest $R^2$ value, suggesting that it can make accurate predictions and explain salary variations well.

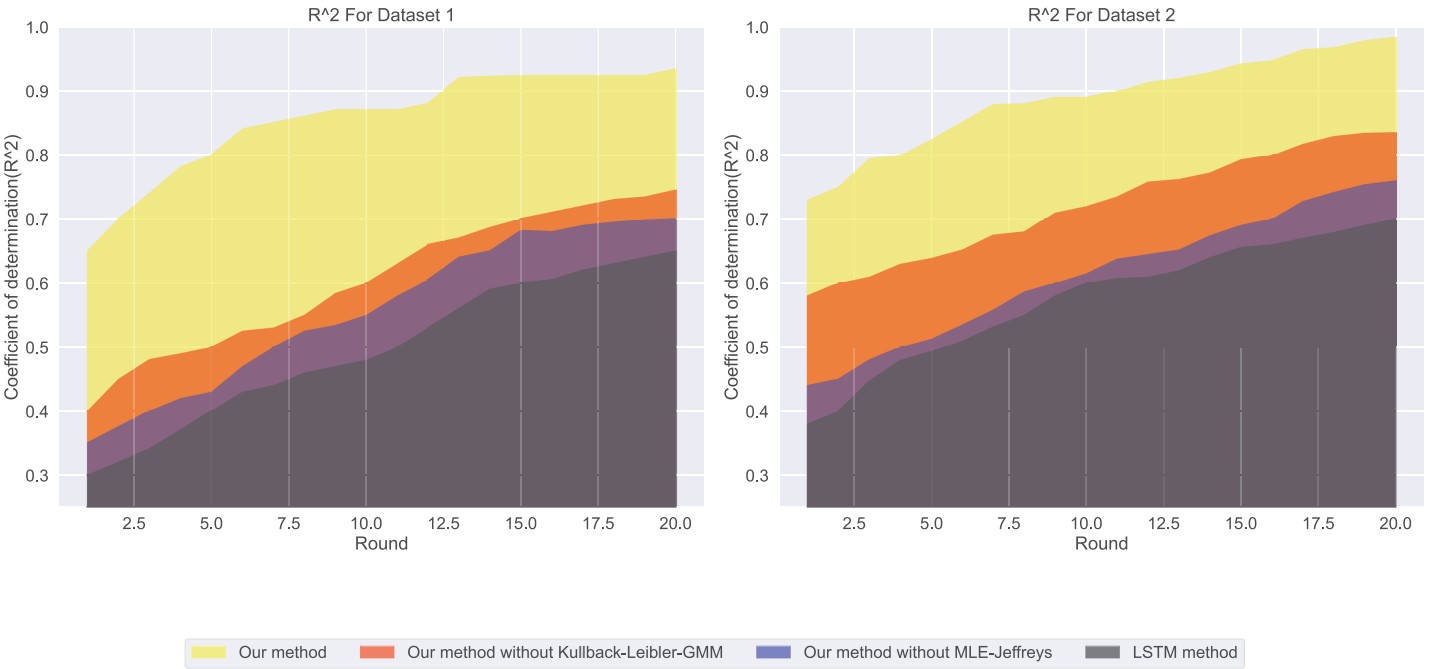

**Figure 2** The coefficient of determination for this experiment: the ($R^2$) value for LSTM methods is the lowest, followed by our method without MLE-Jeffreys, and then our method without Kullback-Leibler-GMM. In contrast, our method has the highest ($R^2$) value, implying that it best explains the variations in salaries.                         

### Quantile loss and evaluation

By comparing four LSTM models, we found that they perform differently in MAE and ($R^2$). To more fully assess the accuracy and risk of forecasting compensation, we introduce two new metrics: Loss and Conditional Value-at-Risk (CVaR). Doing so can help employers, job seekers and researchers more accurately understand salary changes.

Loss is a measure of how close a model's predictions are to actual results, with lower loss values representing more accurate predictions. CVaR is a risk assessment tool used to quantify expected losses at a specific level of confidence. This is especially important for companies to formulate compensation strategies: a high CVaR means a more conservative strategy is needed, while a low CVaR may allow companies to formulate strategies with more confidence.

In the salary prediction experiment, the experimental results are shown in Fig. 3. Among them, our method has the lowest loss, indicating that it has the highest consistency with real data when predicting employment salary. In contrast, LSTM methods have the highest losses, indicating that their predictions are less consistent with the actual data. We then used CVaR to measure the previous model, so we used CVaR to measure it. As shown in Fig. 4, our method has the lowest CVaR value, which indicates that its prediction results have the smallest risk or possibility of loss. On the other hand, LSTM methods have the highest CVaR values, which means their predictions are riskier.

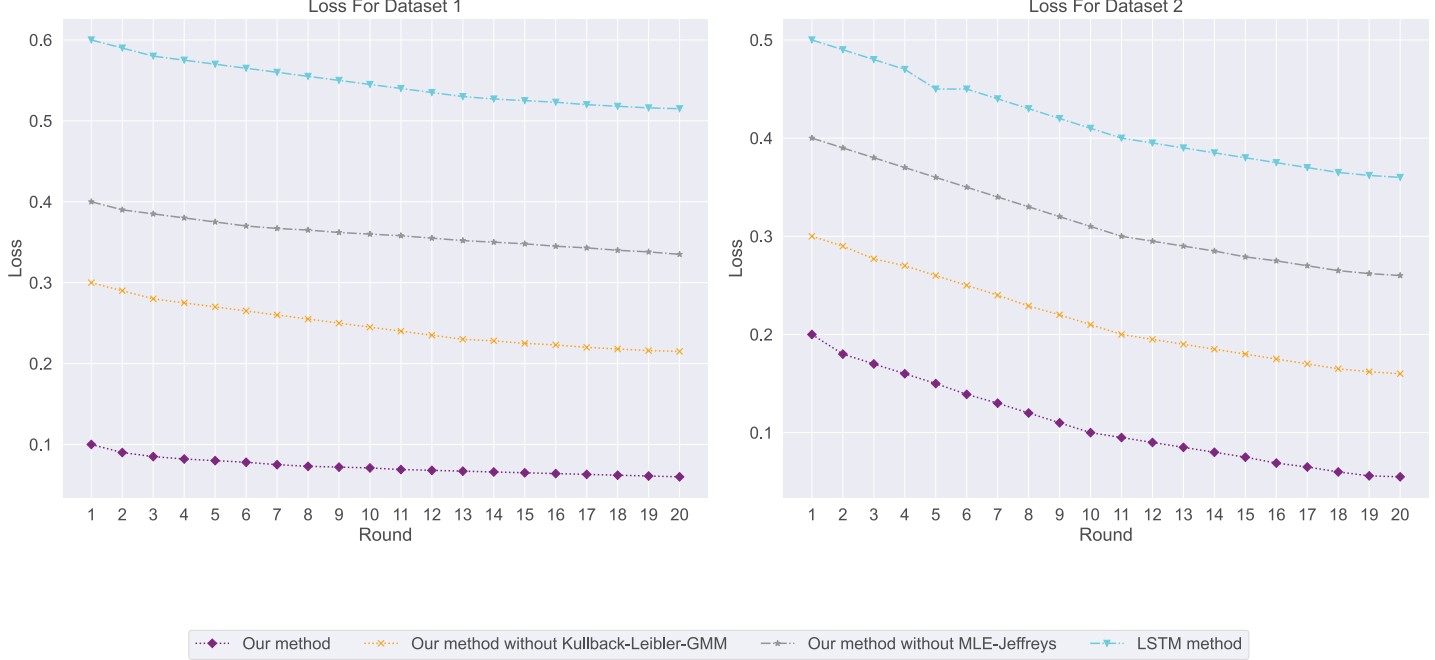

**Figure 3 The loss for this experiment: our method has the lowest loss, followed by our method without Kullback-Leibler-GMM, and then our method without MLE-Jeffreys.** Conversely, the LSTM method exhibits the highest loss, suggesting that our method achieves the highest consistency with the real data when predicting employment salaries.

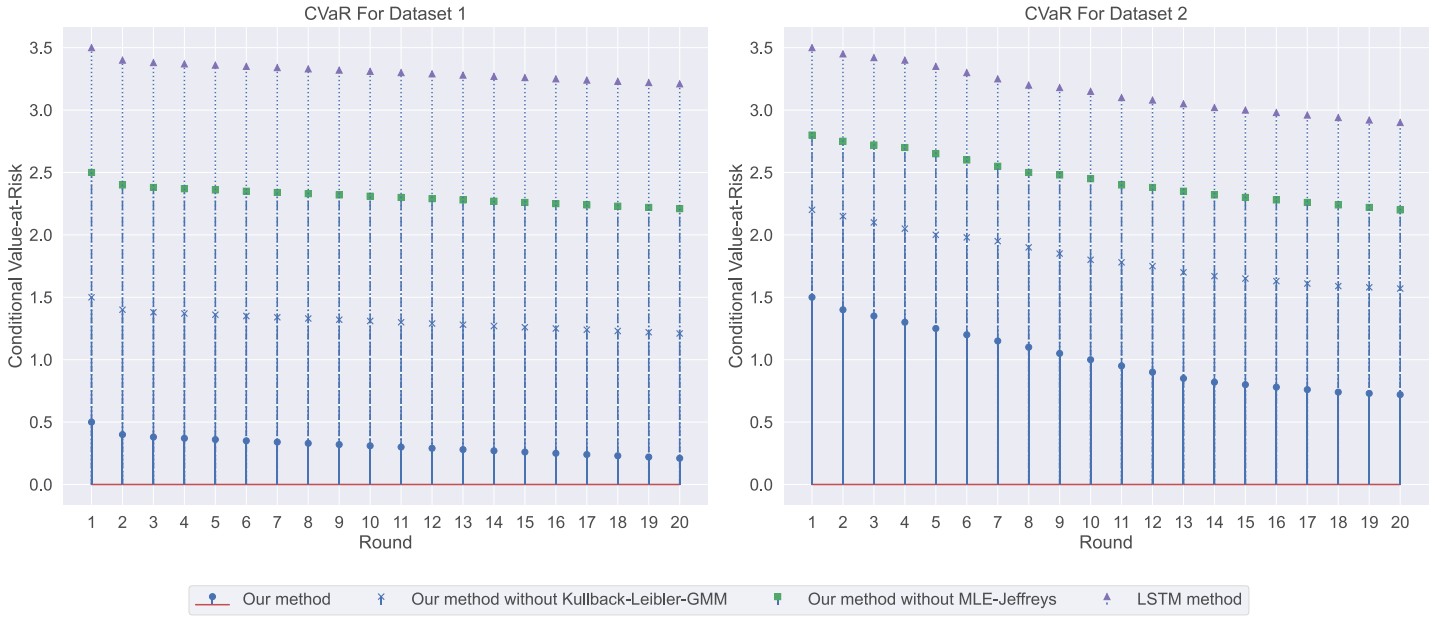

**Figure 4 The conditional value-at-risk for this experiment: our method demonstrates the lowest conditional value-at-risk (CVaR), followed by our method without Kullback-Leibler-GMM, and then our method without MLE-Jeffreys.** On the other hand, the LSTM method has the highest CVaR, indicating that our method has the smallest risk or potential loss in its predictions.

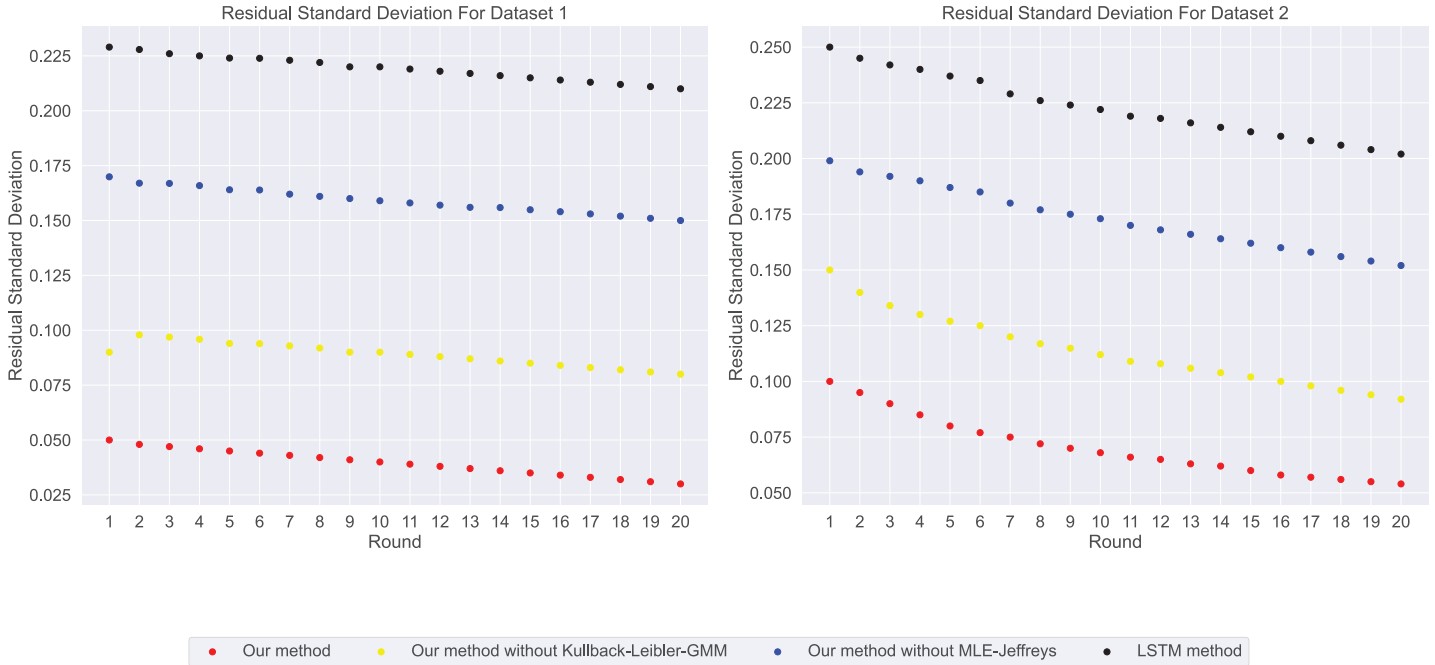

**Figure 5 The residual standard deviation for this experiment: our method shows the lowest residual standard deviation, followed by our method without Kullback-Leibler-GMM, and then our method without MLE-Jeffreys.** Conversely, the LSTM method exhibits the highest loss, indicating that our method maintains relatively consistent and accurate performance across all predictions.

### Stability and generalization metrics

Through in-depth analysis of indicators such as MAE, coefficient of determination, Loss, and CVaR, we recognize that formulating a compensation strategy solely based on prediction accuracy and risk is insufficient. Therefore, we introduce two novel metrics: residual standard deviation and prediction stability. These metrics not only offer profound insights into model errors but also aid in assessing the robustness of predictions. This multi-dimensional evaluation approach enables companies to establish a more comprehensive and flexible compensation strategy to effectively address various challenges. The residual standard deviation serves as a standard measure for prediction errors and accurately reflects the consistency of model predictions. Lower values imply more precise and stable predictions. Prediction stability is used to assess whether the model performs consistently across different scenarios—such as training, validation, and test sets.

In our experimental residual standard deviation results as shown in Fig. 5, our method has the lowest residual standard deviation, which means that it shows relatively consistent and accurate performance in all predictions. On the contrary, LSTM methods have the highest residual standard deviation, suggesting that their prediction results may be scattered and less consistent with real data. The stability results of predicted values in our experiments are shown in Fig. 6. Our method performs well in prediction stability. Its predictions are maintained whether in different data sets or at different time points.

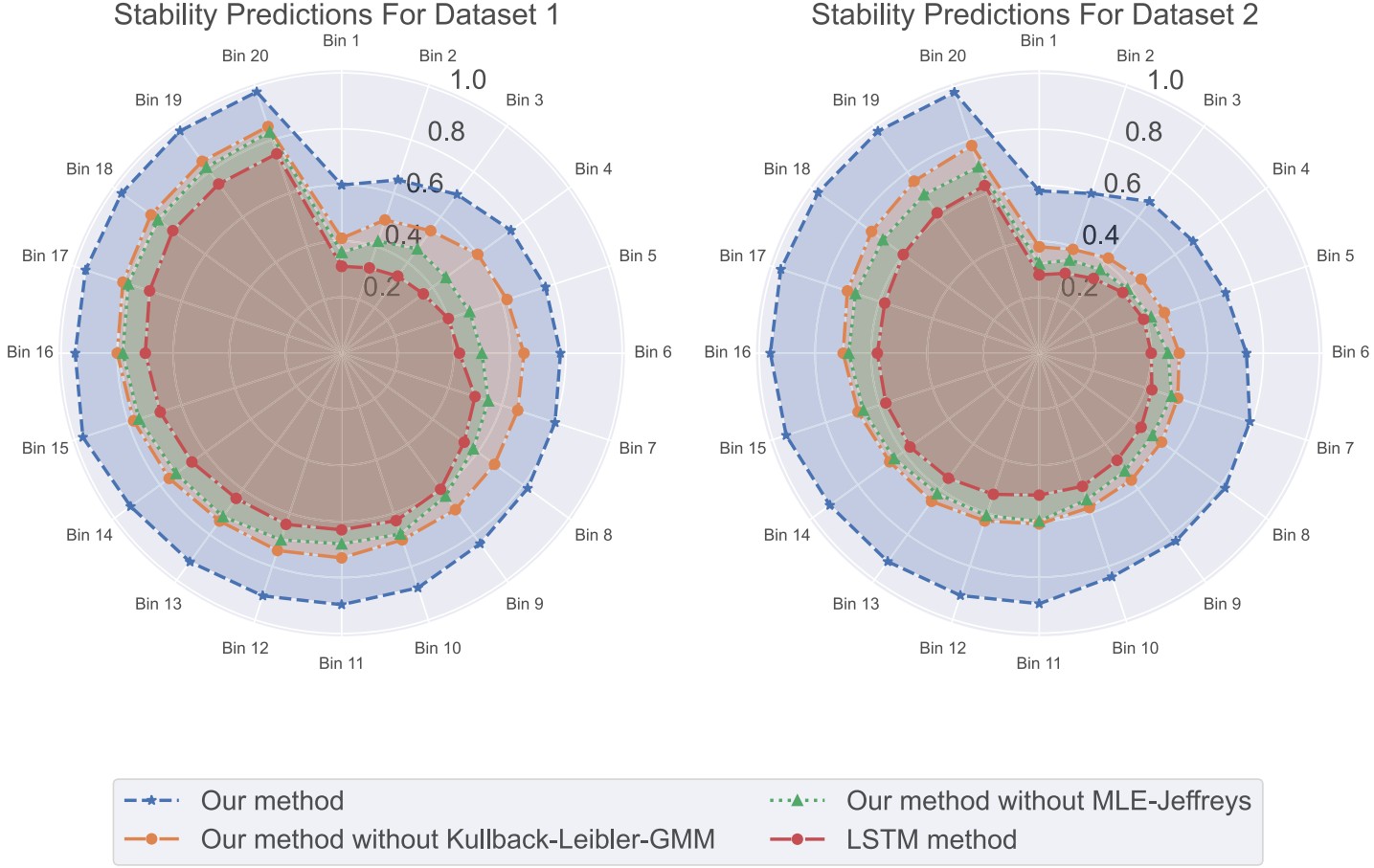

**Figure 6 The stability predictions for this experiment: our method excels in prediction stability, followed by our method without Kullback-Leibler-GMM, and then our method without MLE-Jeffreys.** In contrast, the LSTM method demonstrates poorer prediction stability, whether across different datasets or at different time points. Our method maintains relatively stable performance in its predictions.

**Table 3 Time overhead of different models on two datasets.**

| Time model | Our method -Dataset 1 | Our method without KL -Dataset 1 | Our method without MJ -Dataset 1 | LSTM method -Dataset 1 | Our method -Dataset 2 | Our method without KL -Dataset 2 | Our method without MJ -Dataset 2 | LSTM method -Dataset 2 |
|---|---|---|---|---|---|---|---|---|
| Total run time (s) | 182.2413 | 243.7124 | 307.2156 | 368.9417 | 204.1782 | 263.9125 | 324.8310 | 387.5102 |
| Average time per round (s) | 9.1121 | 12.1856 | 15.3608 | 18.4471 | 10.2089 | 13.1956 | 16.2416 | 19.3755 |

Relatively stable performance. The prediction stability of LSTM methods is poor, which may mean that the model is too complex or overfitted, resulting in inconsistent performance on different data.

### Execution time

In the employment compensation prediction experiment, we have already used multiple evaluation metrics. As shown in Table 3, when using the same model but applying different

algorithms, our method outperforms the other three models in both accuracy and execution time. This phenomenon reveals a series of insights about algorithm selection, performance, and efficiency, and the differences in accuracy and runtime become particularly noteworthy.

Compared to our method, the run time of the other three algorithms is longer, possibly because they are more complex in data processing, requiring more parameters and computational steps. However, it should be noted that increased runtime does not necessarily imply higher accuracy. Our method demonstrates significant advantages in computational time, highlighting not only its algorithmic efficiency but also its ability to converge in a shorter time frame.

## Discussion

Starting salaries after graduation are often considered an important reflection of the quality of university education and the individual efforts of students. Based on multiple studies and data analyses, we have identified several key factors that influence the starting salaries of graduates to varying degrees. First, the choice of major and industry clearly plays a decisive role, especially in STEM (Science, Technology, Engineering, and Mathematics) fields. Secondly, academic performance is another factor that cannot be ignored, as good academic performance is often associated with higher salaries. Furthermore, internships and practical work experience significantly enhance students' competitiveness in the job market.

Therefore, we arrive at the following discussion:

- Our study is based solely on academic performance. We have found in our datasets that in the first dataset, higher exam scores, a stronger pursuit of specialization, higher tier cities of the college, and higher scores in AMCAT English, logical and quantitative abilities are associated with higher salaries. In the second dataset, factors such as higher scholarship types, more basic modes of transportation to university, simpler types of university accommodation, longer weekly study hours, higher frequency of reading, better exam preparation, and stronger classroom listening skills also lead to higher salaries. This range of academic performance indicators demonstrates that our model can use these factors to predict their future salaries and provide guidance and reminders to students on what to do in school to achieve better salaries upon graduation. Our focus extends beyond mere mathematical methods to solving real-world problems, particularly how our mathematical approach can address the practical issue of predicting graduate salaries.
- Our datasets primarily focus on academic performance but do not fully cover aspects like students' economic status, job market trends, and psychological characteristics. The first dataset only partially addresses the economic status and job market trends of some students, such as the AMCAT personality test, conscientiousness, extraversion, openness to experience, and test scores. The second dataset also only partially focuses on aspects such as parental education level, number of siblings, family status, and parents' occupations. Indeed, our datasets do not include students' economic status, job market

trends, and psychological characteristics, but our method still demonstrates effectiveness in predicting salaries through academic performance. This limitation of our study, not incorporating students' economic status, job market trends, and psychological traits, will be addressed in future work. We plan to conduct field research combining students' academic performance with their economic status, job market trends, and psychological traits to comprehensively demonstrate the potential of our method and provide more accurate predictions.

## CONCLUSION

This study provides a comprehensive evaluation of the performance of multiple models in employment salary prediction, including basic error metrics and more complex criteria for risk and stability. Our method performs exceptionally well on all these evaluation standards, demonstrating its outstanding performance not only in prediction accuracy but also in terms of risk and stability. We hope that this study can provide direction and broaden the horizons for future research in this field.

### Funding
The authors received no funding for this work.

### Competing Interests
The authors declare that they have no competing interests.

### Author Contributions
- Fanghong Li conceived and designed the experiments, analyzed the data, performed the computation work, prepared figures and/or tables, authored or reviewed drafts of the article, and approved the final draft.
- Norliza Abdul Majid performed the experiments, analyzed the data, performed the computation work, prepared figures and/or tables, authored or reviewed drafts of the article, and approved the final draft.
- Shuo Ding conceived and designed the experiments, performed the experiments, analyzed the data, authored or reviewed drafts of the article, and approved the final draft.

### Data Availability
The data is available at Kaggle and Salary forecast for engineering graduates:

- CSAFRIT, (2021, October). Higher Education Students Performance Evaluation, Version 1. https://www.kaggle.com/datasets/csafrit2/higher-education-students-performance-evaluation/data.

- Xufengnian, (2021, November). Salary forecast for engineering graduates, Version 1. https://aistudio.baidu.com/datasetdetail/107973.

## Supplemental Information

Supplemental information for this article can be found online at http://dx.doi.org/10.7717/peerj-cs.1875#supplemental-information.

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
