# Peer review of "Unlocking the potential of LSTM for accurate salary prediction with MLE, Jeffreys prior, and advanced risk functions"

_PeerJ Computer Science, doi:10.7717/peerj-cs.1875_

## Round 0.1 · original submission · Major Revisions

The authors must improve the paper accordingly the reviewers suggestions.

**Language Note:** PeerJ staff have identified that the English language needs to be improved. When you prepare your next revision, please either (i) have a colleague who is proficient in English and familiar with the subject matter review your manuscript, or (ii) contact a professional editing service to review your manuscript. PeerJ can provide language editing services - you can contact us at copyediting@peerj.com for pricing (be sure to provide your manuscript number and title). – PeerJ Staff

Reviewer 1 ·

Basic reporting

This paper attempts to introduce a new framework that incorporates MLE, Jeffreys priors KL risk function, and Gaussian mixture to LSTM models to optimize predictions on college graduate starting salaries. In the text, it does mention some prior work in the field, however, it'd be nice to show their model performance (linear regression, random forest or simple lstm) as a comparison. The reason I'm asking for this is because from the given datasets, it does not seem to have many features (less than 20 as far as from dataset introduction under experiments section), I really doubt a single tree based model or even boosting based models will be underperformed. I will need more annotations on the figures too. For example, in figure 1, what are both axes labels? Is y axis the '$'? This is not clear to me. Same for the x axis, I have no idea what the numbers represent. "Traditional statistical methods" was mentioned several places in the text. What is it actually? I really think there're a few sanity checks this paper has to do before introducing their proposed framework.

Experimental design

In terms of experimental design, I do appreciate the fact that the authors included a variety of mathematical techniques as a way of optimization. However, I just don't believe this is the meat of the problem. Regarding how much salaries can be made for college graduates, there were so many factors that can influence the results asides from academics. For example, economic crises, job market downtime and people's personality (people with high confidence might serve as an edge when it comes to looking for a job). I personally believe features are more important than the methods here for this particular problem. Unless authors can show that no other features can add additional predictive power, I'm not convinced completely on the proposed methods otherwise.

Validity of the findings

This study introduces more of a mathematical method rather than trying to solve the actual problem of predicting college graduate salaries. In spite of the comprehensiveness of the methods description, I'm not convinced that this is the main drawback of the problem. Please address the aforementioned comments if you can.

Reviewer 2 ·

Basic reporting

- Clarity of Language: The paper is commendable for its use of clear and unambiguous professional English. The language is well-structured and facilitates easy understanding.
- Notations and Formula Expressions: While the paper is well-structured, the abundance of notations and formula expressions might present a challenge for readers. Despite this, the equations remain generally accurate with only minor errors, contributing to a detailed understanding of the proposed models. The inclusion of complexity analysis and a theorem proof section is appreciated, showcasing a commitment to the theoretical underpinnings of machine learning models.

Experimental design

- Originality and Relevance: The paper aligns with the Aims and Scope of the journal, presenting original research with a well-defined and meaningful research question. It contributes to addressing a knowledge gap in the field.
- Rigorous Investigation: The research design demonstrates a high level of technical and ethical rigor. Methods are adequately described, providing sufficient detail for replication.

Validity of the findings

- Data Availability: All underlying data have been provided, demonstrating robustness, statistical soundness, and control.
- Conclusion: The conclusions are well-stated, connecting back to the original research question and remaining within the boundaries of supporting results.

Additional comments

1. List Formatting (Line 147-153): The attempt to create a list at lines 147-153 needs improvement. The current formatting places all list items in the same row, affecting readability.
2. Hardware Configuration: The transparency regarding hardware configuration in Table 1 is commendable. However, there is a suggestion to increase the font size for better readability.
3. Figures: The figures in the paper are visually appealing, with a clean presentation. However, there is a concern in Section 7.3 where certain axes lack explanation (e.g., x-axis in figure 1-5, circular axis in figure 6). Clarification is needed regarding the range of 1-20; whether it represents training epochs, train-test splits, or another parameter. Additionally, the trend of lines in these figures is unclear and requires explanation.

---

## Round 0.2 · accepted · Accept

The paper is clear and was well improved.

Reviewer 2 ·

Basic reporting

The paper continues to excel in the clarity and precision of its language, maintaining a professional tone throughout. The revised manuscript maintains a well-structured format, and the inclusion of clear notations and formula expressions enhances the understanding of the proposed models. The improvements in list formatting have positively impacted readability. The transparency regarding hardware configuration is commendable, and the suggested increase in font size in Table 1 for better readability has been implemented. Figures in the paper remain visually appealing, with a clean presentation.

Experimental design

The paper continues to present original primary research within the Aims and Scope of the journal. The research question is well-defined, relevant, and meaningful. The author has effectively addressed the motivation for using the Maximum Likelihood Estimation (MLE) model, emphasizing its interpretability and computational convenience. However, further clarification is needed regarding how the claimed oversight of diverse influencing factors in traditional prediction models limits accuracy, as mentioned in the abstract (line 13-17).

The algorithmic explanation for MLE and Jeffreys Prior, as presented in lines 278-291, requires attention, as the line numbers in the pseudocode listing do not align with the description text. Ensuring consistency between the pseudocode and its explanation is crucial for reader comprehension. The sections explaining the motivation for using the MLE model, Jefferys' Prior, KL risk function, and Gaussian Mixture Model (GMM) are appreciated and contribute to a clear understanding of each component in the system.

Validity of the findings

The paper demonstrates robust and statistically sound findings, presenting a noteworthy contribution to the field. The underlying data, known for their robustness and statistical validity, are a strong foundation for future research. The conclusions are thoughtfully crafted, well-aligned with the original research question, and effectively supported by the presented results, reinforcing the overall validity of the findings.

Additional comments

Overall, the paper has made commendable improvements since the previous submission. Attention to the remaining issues highlighted in the comments will further enhance the manuscript's clarity and ensure its alignment with the journal's standards.